# The Dynamic Poly(A) Tail Acts as a Signal Hub in mRNA Metabolism

**DOI:** 10.3390/cells12040572

**Published:** 2023-02-10

**Authors:** Guiying Zhang, Haolin Luo, Xinyi Li, Zhangli Hu, Quan Wang

**Affiliations:** 1Guangdong Technology Research Center for Marine Algal Bioengineering, College of Life Sciences and Oceanography, Shenzhen University, Shenzhen 518055, China; 2Key Laboratory of Optoelectronic Devices and Systems of Ministry of Education and Guangdong Province, College of Optoelectronic Engineering, Shenzhen University, Shenzhen 518060, China; 3School of Pharmacy, Xianning Medical College, Hubei University of Science and Technology, Xianning 437100, China

**Keywords:** dynamic, poly(A) tail, signal hub, mRNA, metabolism

## Abstract

In eukaryotes, mRNA metabolism requires a sophisticated signaling system. Recent studies have suggested that polyadenylate tail may play a vital role in such a system. The poly(A) tail used to be regarded as a common modification at the 3′ end of mRNA, but it is now known to be more than just that. It appears to act as a platform or hub that can be understood in two ways. On the one hand, polyadenylation and deadenylation machinery constantly regulates its dynamic activity; on the other hand, it exhibits the ability to recruit RNA-binding proteins and then interact with diverse factors to send various signals to regulate mRNA metabolism. In this paper, we outline the main complexes that regulate the dynamic activities of poly(A) tails, explain how these complexes participate polyadenylation/deadenylation process and summarize the diverse signals this hub emit. We are trying to make a point that the poly(A) tail can metaphorically act as a “flagman” who is supervised by polyadenylation and deadenylation and sends out signals to regulate the orderly functioning of mRNA metabolism.

## 1. Introduction

Messenger RNA metabolism is one of the most important activities carried out by organisms in response to dynamic environmental and developmental signals. Put simply, it involves mRNA biogenesis, maturation, exportation from the nucleus, translation to proteins, and degradation. These processes require that numerous enzymes and regulatory elements interact with cellular machinery in an orderly manner. A sophisticated signal transmission system is indispensable in such a large and complicated network of interactions. Recent studies have suggested that polyadenylate tails may play a vital role in such signaling systems. Earlier studies concluded that adenosine modifications of mRNA 3′ ends are formed to execute three major functions: to promote mRNA exportation; to maintain mRNA stability; and to increase translation efficiency [1,2,3,4,5,6]. Researchers also developed a vague notion that the poly(A) tail might be a kind of generated signal [7,8]. More recent studies have shown that poly(A) tails are much more complicated than originally thought. Improved poly(A) tail profiles have revealed that highly translated mRNAs possess much shorter poly(A) tails than were previously believed [9,10,11]. The cytoplasmic poly(A)-tail-binding protein (PABPC), which binds to the poly(A) tails and stabilizes the mRNA and itself, has now been shown to stimulate PAN2–PAN3 and CCR4–NOT complex formation, and this mediates the shortening of poly(A) tail lengths and subsequent mRNA degradation [12,13]. These recent findings imply that previous portrayals of poly(A) tails were of limited value, and also strengthen the hypothesis that poly(A) tails mediate interaction among numerous elements involved in mRNA metabolism. Because of this, a new definition and functional understanding of the poly(A) tail is now necessary. In this paper, we articulate our view that the poly(A) tail acts as a signal hub throughout the entire metabolic cycle of mRNA. Metaphorically, it acts as a “flagman” which is proficient in an intricate kind of “semaphore” and is able to deliver diverse signals either by itself or by waving different “flags”. Obviously, these signals are essential to regulating the orderly functioning of mRNA metabolism. In the commentary below, we outline the main complexes that regulate the dynamic activities of poly(A) tails and summarize the diverse signals that poly(A) tails emit, both singly and collaboratively.

## 2. The Dynamic Poly(A) Tail and Its Regulators

Ever since 3′ end polyadenylic acid was discovered in the 1960s, researchers have sought to depict and understand this widespread modification [14,15,16]. Numerous experiments have been carried out, using diverse methods, to determine the features of this adenosine repeat sequence [17,18,19,20]. Decades of research have led to a common understanding among scholars that poly(A) tails at 3′ ends are not decorations set in stone but dynamic entities, constantly regulated by polyadenylation and deadenylation mechanisms throughout the entire mRNA metabolic cycle [21,22].

### 2.1. Nuclear Polyadenylation Regulates Nascent Pre-mRNA Poly(A) Tail Generation

Important regulatory activities associated with the poly(A) tail extension include polyadenylation action in the nucleus (Figure 1A). This usually consists of cleavage and polyadenylation steps. USE (upstream element), PAS (polyadenylation signal), DSE (downstream element), and GRS (G-rich sequence) are four known conserved elements used in synthesizing the poly(A) tail, but numerous complexes such as CPSF (cleavage and polyadenylation specificity factor), CSTF (cleavage stimulation factor), CFIm and CFIIm (mammalian cleavage factors 1 and 2), and HCC (histone pre-mRNA cleavage complex) are also involved in the first step, and then recruit cPAP (canonical Poly(A) polymerase) to fulfill the second step in mammals [23,24,25,26].

After Pol II (polymerase II) passes the PAS, an AAUAAA sequence located 15–30 nucleotide upstream of cleavage site on the 3′ untranslated region, the CPSF complex recognizes and binds to this site [27,28]. CPSF is a complex that contains CPSF160, CPSF30, hFip1 (human Fip1), WDR33, CPSF100, CPSF73, and symplekin (seven subunits). The first four subunits are both necessary and sufficient in themselves; CPSF100, CPSF73, and symplekin are not essential for polyadenylation [29]. Coincidentally, one study showed that these three subunits could form a HCC (histone pre-mRNA cleavage complex) together with CstF64, and this possessed functions equivalent to mammalian cleavage factor (mCF) [30]. A functional differentiation of CPSF subunits might also be noted here, as CPSF73 exhibits endonuclease activity, hFip1 exhibits the ability to recruit poly(A) polymerase, and both CPSF160 and WDR33 form sub-complexes to recruit the remaining members of the CPSF complex [31,32,33,34]. After CPSF binds to PAS, the CSTF complex, composed of the three subunits CSTF77, CSTF64, and CSTF50, recognizes and binds to the GU/U-rich DSE sequence. Although the specific function and architecture of each CSTF subunit is still unclear, their contributions to the promotion of cleavage have been documented [35]. The CFIm complex is composed of CFIm25 and CFIm68/CFIm59, which bind to USE, a U(G/A)UA sequence. It used to be thought that this promoted cleavage, but recent studies have found that it regulates alternative PAS selection through the opposing functioning of CFIm59 and CFIm68 [36,37,38]. Multiple PASs often associate with a single gene. Selected individual PASs have also been shown to be able to create various transcripts in metabolism regulation [39,40,41]. CFIIm is a poorly characterized complex, but it is known to catalyze 3′ end cleavage and polyadenylation. A reconstitution study showed that CFIIm could exist as a heterodimer composed of hPcf11 and two hClp1 subunits bound to newly transcribed mRNA, with nanomolar affinity. The hPcf11 subunit contains a conserved FEGP motif, which is crucial for pre-mRNA cleavage, while hClp1 exhibits RNA 5′ kinase activity, which is not essential for 3′ end processing [42]. Although these complexes are defined in terms of composition and structure, they do not function independently. Frequent and complicated interactions exist not only between the complexes themselves, but also among subunits within or across complexes. 

Interactions among the above complexes and elements result in pre-mRNA 3′ breakage and cPAP aggregation at the cleavage site, as well as initiation of the adenosine residue addition to the 3′ end. This is actually the second step of nuclear polyadenylation. A group of poly(A) polymerase, non-template-dependent RNA polymerases which specifically utilize ATP (adenosine triphosphate) as a substrate to create polyadenylate tails under Mg2+ circumstances, is responsible for this part of the work. Canonical PAPs specifically function in the nucleus and co-transcriptionally interact. In humans, these include PAPα, PAPβ, and PAPγ [43,44,45]. Along with the extension of the poly(A) tail, the PABPNs (nuclear poly(A)-binding proteins) bind to the newly formed adenylate tail in charge of tail-length limitation and mRNA exportation [46]. Some other studies have suggested that PABPN binding may regulate PAS selection and pre-mRNA splicing; however, the specific mechanism involved remains unclear [47,48]. What is certain is that PABPN binding and mRNA exportation mark the termination of the nuclear polyadenylation process. 

### 2.2. Cytoplasmic Deadenylation Regulates Poly(A) Tail Shrinkage and mRNA Decay 

Nuclear polyadenylation promotes poly(A) tail extension. In contrast, cytoplasmic deadenylation contributes to poly(A) tail shrinkage (Figure 1A). Exactly how initiation occurs remains unclear, and the particular mRNA tail which undergoes deadenylation is also not yet known. PAN2–PAN3, CCR4–NOT, and PABPC are all widely accepted as playing vital roles in mediating poly(A) deadenylation [49,50,51]. PAN2–PAN3 is a heterotrimer composed of a PAN2 subunit and an asymmetric homodimer of two PAN3 subunits, which may be preferentially recruited to a long poly(A) tail that is ready for deadenylation [52,53]. PAN2 consists of an N-terminal W40 domain, a C-terminal UCH (inactive ubiquitin C-terminal hydrolase), and exonuclease domains. The N-terminal and C-terminal domains are linked by a long PID (PAN3-interacting domain). PAN3 contains a ZnF (zinc finger) in its N-terminus; a PKC unit that consists of a pseudokinase, a coiled coil, and a C-terminal domain (CTD) together; and a PAM2 (PABP-interacting motif 2) in between. PAN3 exhibits no deadenylation activity, but it provides a link between PAN2, PABPC, and adenosine [52,54,55]. Specifically, it interacts with the PID of PAN2 through its C-terminus, while binding to the adenosine tail with the N-terminal zinc finger region. A strong interaction between PAN3 and PABPC has been identified in yeast [56,57]. PAN2–PAN3 deadenylation activity, which is regulated by phosphorylation, may also be promoted by PABPC [58,59]. The WD40 domain on PAN2 has the capacity to promote protein interactions [60,61]. These findings imply a complicated and subtle interaction network comprising the PAN2–PAN3 complex, poly(A) RNPs (ribonucleoprotein complexes of poly(A)-binding proteins), and other factors which co-operatively promote deadenylation [12,40]. 

CCR4–NOT is a large protein complex which exerts a broad range of effects upon mRNA metabolism [62,63]. An early study revealed that, in yeast, the CCR4–NOT complex consists of at least nine subunits, including CCR4 (C-C chemokine receptor type 4), CAF1 (CCR4-associated factor 1), NOT1 (negative regulator of transcription subunit 1), NOT2 (negative regulator of transcription subunit 2), NOT3 (negative regulator of transcription subunit 3), NOT4 (negative regulator of transcription subunit 4), NOT5 (negative regulator of transcription subunit 5), CAF40 (CCR4-associated factor 40), and CAF130 (CCR4-associated factor 130) [64]. Among these, CAF130 is found only in yeast, while the other subunit homologs are found in most eukaryotes. Generally, the CCR4–NOT complex and its subunits can be classified into four modules. NOT1, which is the largest subunit of the CCR4–NOT complex, forms a scaffold which enables other subunits to combine [65]. NOT2, NOT3, and NOT5 together form a NOT module whose function remains unknown [66]. The deadenylase modules of CCR4–NOT are composed of CCR4 and CAF1 subunits [67]. Although both CCR4 and CAF1 catalyze adenosine removal from the 3′ ends, they are applied in different situations. CCR4 catalyzes PABPC-bound poly(A) tail deadenylate, while CAF1 catalyzes naked poly(A) tail deadenylate. CAF1 activity is usually suppressed by PABPC; it can be activated only when PABPC falls off [51]. The ubiquitin module of CCR4–NOT is mediated by NOT4 subunits next to the deadenylase module [68]. CAF40 binds to NOT4 and may contribute to miRNA-mediated RNA degradation [69]. The fourth module of CCR4–NOT is a nonenzymatic module identified in humans which is composed of CNOT10 (human CCR4–NOT transcription complex subunit 10) and CNOT11 (human CCR4–NOT transcription complex subunit 11) [61,70].

Although there are some indications that CCR4–NOT mediates nuclear deadenylation of poly(A) tails after the induction of a serum response, no universal effect has yet been confirmed [71]. PAN2–PAN3 and CCR4–NOT are mainly found localized in the cytoplasm, and these are responsible for most poly(A) tail deadenylation and subsequent mRNA degradation [72].

### 2.3. Oocyte Cytoplasmic Poly/De-Adenylation Regulates the Poly(A) Tail to Replenish Transcription 

The third regulatory mechanism that affects the poly(A) tail is the oocyte cytoplasmic poly/de-adenylation (Figure 1B) observed in the oocytes of many species [73,74,75,76,77]. Due to the absence of transcription in this particular developmental period, the need for a more highly expressed regulatory policy and the reuse of already translated mRNA can be readily understood. To this end, a flexible cytoplasmic poly/de-adenylation mechanism evolved [78]. In order to meet different needs in addition to oocyte development, this process can be repeatedly inhibited and reactivated [77,79,80]. Although this mechanism is only observed in oocytes, it may represent the most intricate process of poly(A) tail regulation. Many factors and elements have been suggested as constituent parts of this temporary poly(A) regulation process, but only a few of these have been considered in any detail to date. 

The authors of [81] investigated *Xenopus* oocyte maturation and cytoplasmic polyadenylation mediated by CPE (cytoplasmic polyadenylation element) and CPEB (CPE binding protein). In this case, the U-rich CPE sequence, containing at least four U residues present as UUUUAU or UUUUAAU, binds by CPEB [81,82]. As in nuclear polyadenylation, the cytoplasmic CPSF complex binds to the PAS site as well. However, this cytoplasmic CPSF complex differs from its counterparts in the nucleus as the endonuclease CPSF3 (CPSF73) subunit is absent, and no test has yet determined whether the Fip1 subunit is present or not [83]. The noncanonical poly(A) polymerases GLD2 and-possibly-GLD4 might be recruited through interaction with CPEB and CPSF to re-extend the poly(A) tail [76,84]. However, the polyadenylate activities of GLD2 and GLD4 are usually suppressed by PARN (poly(A)-specific ribonuclease), which interacts with CPEB [85]. This suppression can also be achieved by phosphorylation of CPEB during oocyte development [80]. In addition, the PBE (Pumilio-binding element) located at the 3′ UTR, and its binding protein PUM (Pumilio), are other factors involved in cytoplasmic polyadenylation [86,87,88]. PUM not only interacts with CPEB to stabilize its binding to CPE, but also interacts with CCR4–NOT, which is now known to be an important catalytic deadenylase complex [89]. These findings are summarized in Figure 1B, which illustrates a cytoplasmic polyadenylation-inhibition-deadenylation model for oocytes.

### 2.4. Novel Cytoplasmic Polyadenylation Operate Transcript during Early Embryogenesis

Recent studies have reported that DICER-2, an RNAi-related endoribonuclease, promotes cytoplasmic polyadenylation (Figure 1C) in early-stage embryos of *Drosophila* in a CPEB-independent manner [90,91]. Two co-factors of DICER-2 that mediate cytoplasmic polyadenylation are ATX2 (Ataxin-2) and TYF (Twenty-four); these have been found to stimulate clock protein PERIOD synthesis with the assistance of PABP [92,93]. In addition, WISPY, a poly(A) polymerase GLD2 homolog B in *Drosophila*, exhibits polyadenylate activity, which indicates an alternative mechanism for mediation of poly(A) tail elongation [94]. This DICER-2-mediated mechanism is estimated to target 50–60% of mRNAs and thus has the potential to regulate their cytoplasmic polyadenylation. Although there have been few studies on the mechanism of DICER-2, a preliminary model is shown in Figure 1.

## 3. “Signals” Poly(A) Hub Transmission with/without “Flags”

The generation of poly(A) tails via nuclear polyadenylation involves a process which is reminiscent of “flagmen” serving on ships within a fleet. Similarly, the cytoplasmic deadenylation and polyadenylation processes can be thought of as “commanders” who supervise the poly(A) tail dynamics. It should now be clear that the dynamic poly(A) tail is not involved with transcription efficiency or with mRNA stability, as many researchers originally thought [9,95,96,97]. Instead, the dynamic poly(A) tail acts as a platform or a hub, recruiting binding proteins or other factors for diverse regulatory functions (Table 1).

### 3.1. Nuclear Poly(A) Tails Are mRNA Export Signals Controlled by Specific Length and PAPBN Binding

Following transcription and processing in the nucleus, the vast majority of RNAs are exported to the cytoplasm. It is now clear that different types of RNAs, including rRNAs, tRNAs, mRNAs, and snRNAs, utilize specific-and, in some cases, partially overlapping-nuclear exit pathways [98]. That is, the cellular export machinery must be capable of discriminating between distinct RNA species, and each RNA species should possess identifying characteristics that specify their export pathway. The poly(A) tail could be a recognition signal for mRNA nuclear exportation [99].

First, the poly(A) tail may increase the length of the mRNA, which plays an important role in the selection of the RNA export pathway to be used [100,101]. The export behaviors of U-snRNA and mRNA exchange involve length change. U-snRNA would behave like an mRNA in its export if the U-snRNA was elongated by insertion of various RNA sequences of >300 nt. On the other hand, the mRNA would behave like a U-snRNA in its export if the mRNAs were shortened to ∼130 nt or less [100]. Thus, the presence of a poly(A) tail is more of a signal to the cell than it is to the mRNA to help select the mRNA output channel.

Second, the poly(A) tail is thought to be bound to poly(A)-binding proteins that are predicted to function in the recruitment of exportation factors [102]. The proposal that mRNA 3′ end processing and nuclear exportation are closely coupled in vivo is supported by an increasing body of evidence [103,104]. Hector et al. found that Nap2p was independently required for the regulation of the polyadenylation of mRNAs and their subsequent nuclear export [6]. In the case of mRNA exportation, Nab2p associates with transcripts during the formation of the 3′ end and subsequently interacts with an NCP (nuclear pore complex)-associated factor to form a complex that promotes targeting of messenger ribonucleoprotein particles (mRNPs) to the NCP, in a similar way to the functioning of the Mex67p–Mtr2p complex for transcribed mRNA regions [105,106].

Although poly(A) sequences may induce mRNA export, in cases where the poly(A) tail is very long, the transcript is not translocated to the cytoplasm, but instead is retained in the nucleus [6,99]. This is a notable observation which suggests a checkpoint mechanism that controls mRNA quality by blocking or allowing their nuclear export [107,108]. We may therefore speculate that the proper length of the poly(A) tail is an important signal for nuclear export. Although the precise reason for this requires further investigation, we may suggest that this helps to explain the relatively conservative initial mRNA length in each species. Mammalian poly(A) tails extend to a maximum of 200–300 nucleotides, and the nuclear poly(A)-tail-binding protein PABPN1 (formerly PABP2) is critical for regulating the extent and efficiency of polyadenylation [109,110]. Furthermore, the synthesis of poly(A) tails of around 250 nt in length has been demonstrated by radioactive labeling of newly transcribed RNAs in mammalian cell culture systems [2,111]. In *Saccharomyces cerevisiae* mRNA, the poly(A) tails are considerably shorter (70–90 nucleotides), and the nuclear poly(A)-binding protein Nab2 is required for limiting poly(A) tail length [6,112,113]. These findings lead us to speculate that each species adopts different ways of keeping the initial length of the mRNA constant, and these may play a role in signal recognition.

### 3.2. Cytoplasmic Poly(A) Tails Provide Diverse Signals through Binding PABPs

PABPs (poly(A)-binding proteins) are highly conserved RNA-binding proteins in eukaryotic species. Specifically, they recognize the polyadenylated sequence at the 3′ end of the mRNA of eukaryotes. At least three important signals are provided by the combination of PABPs and the poly(A) tail, and these are involved in the regulation of mRNAs and their eventual fate. The first signaling event is the formation of the cap–eIF4E–eIF4G–PABPC-poly(A) complex, which regulates the interaction between the mRNA and the translation machinery. PABP may use both of its initial RRMs (RNA-recognition motifs) to interact with the initiation factor eIF4G [114] and with the translation initiation factor 4E (eIF4E), a 5′-cap-binding recognition protein. The eIF4G:eIF4E complex interacts more transiently with the ATP-dependent RNA helicase eIF4A to form an eIF4F complex [115]. Then, the protein eIF4G and eIF4F complex acts as a scaffold that brings other members of this complex into association [116] and forms a “closed loop”, which is thought to stimulate translation [117]. The second signal is the formation of the PAN2–PAN3–Pab1–poly(A) tail complex, which is involved in the regulation of the interaction between the mRNA and the deadenylase machinery. The poly(A) tail recruits PAN2–PAN3 through at least three interactions: between the PAN3 N-terminal zinc finger and adenosines in the poly(A) tail; between PAN3 and a PAM2 motif in PABP; and between additional regions of the PAN2–PAN3 complex and the PABPC–poly(A) RNP [12,52,118]. As a consequence, the PAN2–PAN3–Pab1–poly(A) tail complex stimulates deadenylation, which has been observed both in vivo and-using fully purified components—in vitro [52,119]. The third signaling event is the formation of the CCR4–NOT–TOB complex. The CCR4–NOT complex is recruited indirectly through the protein adapters TOB1/2m, either one of which associates with PABP [120,121]. Yi et al. demonstrated that the CCR4–NOT complex is a generic deadenylase and that its two catalytic subunits have distinct activities on PABPC. CAF1 degrades naked poly(A) RNA and is blocked by PABPC, whereas CCR4 is able to release PABPC to deadenylate PABPC-bound poly(A) RNA [51].

The mechanism by which the poly(A) tail mediates the formation of the three important signals in the cytoplasm may involve still more factors, and the specific mode of regulation requires further investigation. It is possible that the initial mRNA that enters the cytoplasm has a few unknown markers (such as the proportion of PAPBN and PABPC) [122], and so is more involved in forming the “closed loop” and initiating translation. As translation progresses, the mRNA and various factors bound to it change, at which point the mRNA promotes the formation of the PAPBC–PAN2–PAN3–poly(A) complex and starts to undergo deadenylation. It remains unclear how PAN2–PAN3 activity is regulated, but its deadenylation activity seems to be related to the length of the poly(A) tail [53]. The widely accepted “biphasic deadenylation” model indicates that PAN2–PAN3 removes the distal part of the poly(A) tail, i.e., the longer poly(A) tails, while CCR4–NOT removes adenosines that are closer to the 3′ UTR, i.e., the shorter poly(A) tails. In mammals, the decay of mRNA starts with PAN2–PAN3 complex formation, which mediates shortening of the poly(A) tail to ~110 nt [123]. In yeast, tail length falls to ~40 nt [53] and the CCR4–NOT complex then completes the deadenylation of poly(A) tails to a length of ~25 nt [51]. By such means, the poly(A) tail and PABPs together make up a signaling system to determine the fate of the mRNA.

### 3.3. Poly(A) Tails Provide Degradation Signals by Means of Shortening, Uridylation, and Interaction with Various Factors

Following deadenylation, the poly(A) tails of the mRNA have a length of less than ~25 nt, which is insufficient for binding to PABPs. Consequently, PABPs are released from the mRNA [51]. In general, these short exposed poly(A) tails will gain a U tail through the redundant action of TUT4 and TUT7, and this functions as a degradative signal [124]. Uridylation preferentially occurs on shortened poly(A) tails in plants and animals, presumably from the antagonistic effects of PABPC1 with TUT4 and with TUT7 [125,126]. The oligo-U tail triggers degradation by serving as a signal that is recognized by downstream factors, for example, the LSM1–7 complex and the 3′–5′ exonuclease Dis3L2 [102,124]. The LSM1–7 complex can associate with oligo-U tails or oligo-A tails to help recruit the decapping machinery and then contribute to decapping itself, with the resulting DCP1/2 complex [127,128]. Decapped mRNAs are degraded in the 5′–3′ direction by XRN1 (5′–3′ exoribonuclease 1) [124,129]. Alternatively, the U tail is recognized by exosomes or Dis3L2, which degrade mRNA via an exonucleolytic process from the 3′ end [102,124]. It is currently unclear whether deadenylated mRNAs are degraded via alternative pathways independent of uridylation, and what fractions are involved if they are. In mammals, the oligo-U-tail pathway could be involved in other degradative and surveillance pathways, and might also play a general signaling role, analogous to miRNA-mediated gene silencing [124]. The short mixed tail typically carries the signal of degradation, but the situation is different for different species. In the filamentous fungus *Aspergillus nidulans*, mRNAs carry 3′ tails mixed with cytidine and uridine, and these both serve as degradative signals despite the difference in their base composition [130,131].

### 3.4. Poly(A) Tail Shrinkage Rate May Provide Another Signal

Deadenylation and decay rates have previously been studied only as derivations of studies of mRNA deadenylation and degradation dynamics [51,132,133,134]. Recent studies, however, have demonstrated that the deadenylation and decay rate constants, for which the ranges are numbered in thousands, are much greater than expected, and correspond to cytoplasmic lifetimes [13]. We therefore speculate that the rates of deadenylation and decay may serve as a signaling mode in cells, and that they are broadly involved in mRNA metabolism. This matter has not been adequately investigated by researchers before now. Previous studies have consistently suggested that mRNA stability is related to poly tail length, but the most recent data show that the shortening of poly(A) tails to lengths of about 30 nt can help to maintain mRNA stability [9]. The half-life of mRNA is correlated with the deadenylation rate rather than with tail length, we contend that the rate of deadenylation plays an important role in regulating the rate of translation [9,13]. In this regard, it is interesting to note that the rate of deadenylation is related to the rate of subsequent mRNA degradation. For mRNAs with faster degradation rates, the prior rate of deadenylation is also faster [13]. Overall, the rates of deadenylation and mRNA degradation are closely linked to mRNA metabolism. The mechanisms by which this rate-changing signal is recognized by cells and the means by which it regulates mRNA metabolism deserve further investigation.

**Table 1 cells-12-00572-t001:** Signals provided by poly(A) tail hub, singly and collaboratively.

Location	Species	Flagman	Flags	Signal	References
		Poly(A) Tail Length	Poly(A)–Binding Proteins		
Nucleus	Human	12 nt	PABPN1	stimulates processive poly(A) tail synthesis	[97]
		~250 nt	PABPN1	determine the tail length and assist in mRNA transportation to the cytoplasm	[2,111]
	Yeast	70–80 nt	Nab2	limiting poly(A) tail length and Connecting polyadenylation to RNA export	[6,112,113]
Cytoplasm	Most of eukaryotic	>30 nt	PABPs	stimulate translation or deadenylation, preventing precocious uridylation and decay	[9,51,116]
	Mammalian cells	200–110 nt	PAPBC1	initial phase of deadenylation	[51,53,123]
	Mammalian cells	110–25 nt	PAPBC1	completes deadenylation	[123]
	Yeast	>60 nt	two or more Pab1s	initial phase of deadenylation	[12,51]
	Yeast	20–40 nt	Pab1s	completes deadenylation	[53]
	Most of eukaryotic	naked poly(A)	CAF1	degrades naked poly(A)	[57]
	Most of eukaryotic	<25 nt	TUT4/7	Uridylation–dependent decay	[51,124]
	*Aspergillus nidulands*	CUCU tail		decay	[130,131]

## 4. Conclusions

The eukaryotic poly(A) tail is a general mRNA modification which has been observed for more than half a century. Many researchers have argued that it might be a transcriptional regulatory signal; however, a growing body of evidence has revealed that it also exhibits the ability of binding and recruiting diverse factors. Advanced detection techniques, detailed poly(A) tail profiles and in-depth analyses have shown that the poly(A) tail structure and its binding factors, as well as other subrecruited components, may collectively form a sophisticated signal center within which diverse messages mediate orderly mRNA metabolism. In short, the poly(A) tail acts more like a dynamic signal hub than as an independent signal, while its own dynamic activity is regulated by polyadenylation and deadenylation mechanisms.

## Figures and Tables

**Figure 1 cells-12-00572-f001:**
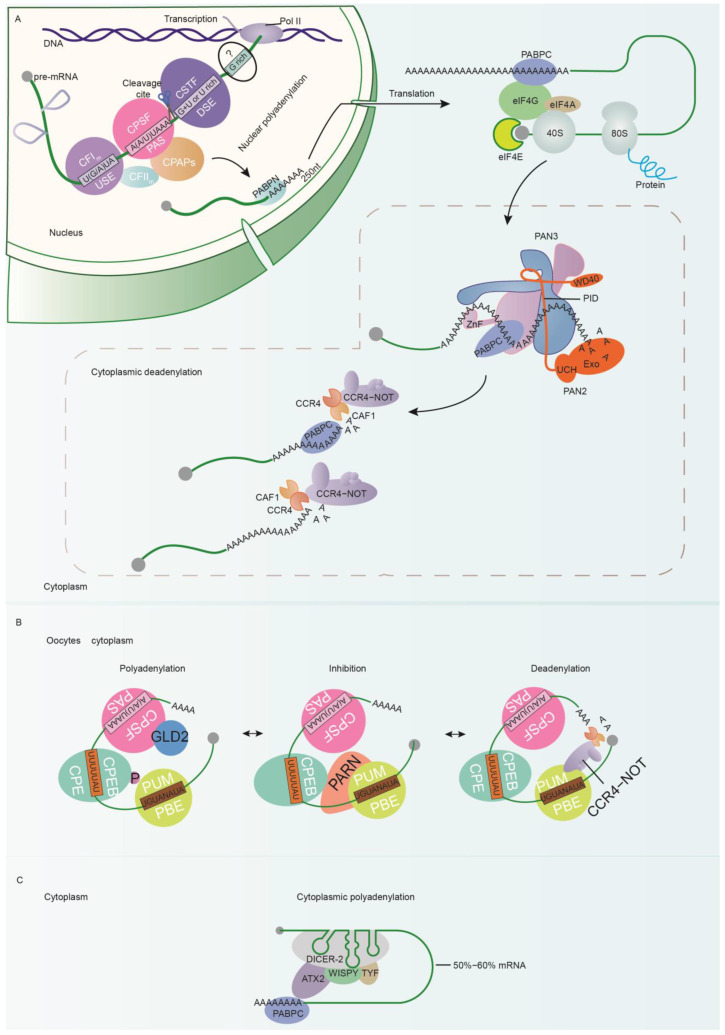
Schematic diagram of polyadenylation and deadenylation mechanisms. (**A**) In the nucleus, when nascent pre-mRNA is transcribed by Pol II, CPSF, CSTF, and CFIm complexes bind to PAS-, DSE-, and USE-specific elements, respectively, and also recruit additional participants such as CFIIm, which jointly promote RNA cleavage at a particular site. Following cleavage, cPAP (canonical Poly(A) polymerase) binds to this site and starts polyadenylation. PABPN then binds to the newly synthesized poly(A) tail for the twin purposes of limiting tail length and releasing mRNA export signals. When mRNA is transported to the cytoplasm, the poly(A) tail and binding protein PAPBC release signals that promote ribosomal machinery translation. The cytoplasmic deadenylation mechanism is then activated, mediating adenosine dissociation and subsequent mRNA degradation. In this process, the PAN2–PAN3 complex preferentially acts on transcripts with long poly(A) tails and PABPC binds. The PAN3 subunit promotes interaction among PAPBC, PAN2, and the poly(A) tail. The Exo (exonuclease) domain on the C-terminal of PAN2 catalyzes the shortening of the poly(A) tail. The CCR4–NOT complex tends to recognize the slightly shorter poly(A) tails. CCR4–NOT contains CCR4 and CAF1, two subunits which form a deadenylase module. CCR4 catalyzes PABPC-bound poly(A) tail deadenylate, and CAF1 catalyzes naked poly(A) tail deadenylate. mRNA degradation then begins on poly(A)-tail-dissociated mRNAs. (**B**) Flexible poly/de-adenylation mechanism in oocyte. In one of these, CPEB and CPSF bind to CPE and the PAS site, and subsequently recruit GLD2. GLD2 can catalyze a relengthening of the poly(A) tail. However, GLD2 activity can be suppressed by PARN. PARN can interact with CPEB and PUM, which bind to the PBE element. Suppression of PARN can be alleviated by CPEB phosphorylation. In addition, PUM can recruit the CCR4–NOT complex which mediates adenosine dissociates. Thus, a polyadenylationinhibitiondeadenylation model is formed. (**C**) A novel cytoplasmic polyadenylation mechanism. RNAi-related endoribonuclease DICER-2 and two co-factors, ATX2 and TYF, together recruit WISPY (GLD2 homologous), leading to polyadenylation of 50–60% of oocyte mRNA.

## Data Availability

Not applicable.

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
