# Peer review of "The Dynamic Poly(A) Tail Acts as a Signal Hub in mRNA Metabolism"

_cells, 2023, doi:10.3390/cells12040572_

Round 1

Reviewer 1 Report

Zhang and others present an “Opinion” manuscript on the function of 3' poly(A) in eukaryotic organisms. It is an overall nice topic to provide a review on, despite many other existing works. I do not think concepts presented are novel enough to justify an “opinion” type of manuscript, but, in principle, the manuscript can be worked into a nice mini-review/summary type of publication. It did not seem reasonable to provide minor feedback at this stage of the manuscript development, thus I mostly limit my comments to some major points which, in my opinion, must be fixed in a re-submission or a major revision. Most of these are related to language and a non-scientific use of such.

Introduction directly repeats statements from the abstract, could this be corrected. I do not like the “flagman” metaphor at all, I think it is inaccurate and unnecessary. It complicates the meaning rather than clarifies anything. The expression used in the title, “signal hub”, is probably already a sufficiently abstract metaphor that does not need any further inflating.

I could not understand sentences such as “Automatically, a vague notion of that poly (A) tail might be a kind of signal generated” (lines 39-40) at all.

I could not understand sentences such as “Thus, a novel and more vivid definition and functional cognition of poly (A) tail are necessary and meaningful.” (lines 48-49).

Figure 1 needs to be re-structured to optimise the font and better depict interactions between the molecules (content is disproportional). While some of the components of the respective polyadenylation processing complexes are depicted, from the figure it is not at all clear how these are regulated and what is their function and mechanism of operation. My suggestion is to show types of regulation in Figure 1 and then, in dedicated figures n the respective sub-sections, show each of these in detail. Otherwise 2.1 and 2.2 are not appropriately illustrated and given their format of (mostly) continuous block of text, are not engaging to follow.

Lines 57-58, I disagree 3’ poly(A) is a modification of RNA. It is a post-transcriptional and also non-templated addition. Also, from the sentence it is not obvious that the 3' poly(A) is meant. Polyadenylic acid may have no relevance to mRNA, or poly(A) stretches can be found in the other locations of mRNA – something that is actually interesting but not discussed in the manuscript. A couple of works where poly(A) has been shown or suspected to be functional elsewhere (not in the 3' end): PMIDs 17761883, 31510048, 27437580.

Line 89, I could not understand the sub-section title. Line 99, “PAS” not explained sufficiently clear.

Lines 129137, PAPs are introduced very abruptly. Overall subsection structure and segregation into the paragraphs is cumbersome. What is the system behind paragraphs?

Line 139, perhaps “Cytoplasmic deadenylation can reduce 3' poly (A) length and facilitate mRNA decay”?

Line 140, I could not understand the opening sentence at all.

Line 161, has not CCR4-NOT been already introduced in the preceding section? Why it is described in detail here?

Overall, sub-section 2.3 follows types of complexes as the base for the paragraph structure, which is different to the other sub-sections (mostly looking at the timeline or types of processing). This needs to be completely reconsidered and restructured, to provide a uniform flow. The structure can be based on each of these categories, but not on mixed types. Mixing categories creates a convoluted and poorly systematised presentation.

Line 225, I cannot understand Section 3 title. Perhaps “General signalling by mRNA 3’ poly(A)”?

Table 1 can be quite useful, but the “signal” is often mixed up with “condition” or “circumstance”. Perhaps, altering definition to more generic (e.g. “effect”) or introducing more columns/types could help resolving this. It is not clear from the table if the “poly(A) binding proteins” contribute to the effects/signals directly? Perhaps not, but then what is the mechanism? Needs to be added. Also, it would be good to cite the respective key references. I do not like the classification into species as it is depicted. Could this be more systematic/same for nuclear and cytoplasmic processes? Information such as in the table, with the specific effects of length/species, could benefit of also being depicted in an illustration.

Material in 3.2 and 3.3 must be reflected by respective illustrations.

Lines 352-353, “Since translation speed is directly linked to mRNA stability” – how is that? What is translation speed? I do not think there is a simple and uniform link of mRNA stability and any efficiency of translation. Also, for such a bold statement, at least some supporting citations would be necessary. This is repeated in lines 352-353 “Since translation speed is directly linked to mRNA stability”, I do not agree and even if in some cases this is so please provide references. Further, I cannot see the conclusion the authors are making towards the rate of deadenylation reducing translational efficiency. Of course, it can reduce it through the shorter 3' poly(A), but this is an effect of 3' poly(A) length, not deadenyltion rate. I think answering this question would require setting up quite specialised experiments.

I do not follow the logic in lines 343-348.

Authors do not cite many recent works on the topic, and I think the breadth of the grasp of the literature is insufficient. This includes recent excellent reviews on the topic. Examples: PMIDs 33522422, 33830517, 34400637, 34385261, 34385261, 30006991, 32807991, 28986506.

Manuscript text needs to be re-worked and made more concise. Already starting from the opening sentences in the abstract, there is a great deal of repetition in the statements and wordiness. Different concepts/categories are mixed at the same level. Grammar often makes it impossible to understand the exact meaning of a sentence. Authors use far too many unnecessary “mood” adjectives, such as “obviously”, “basically”, “automatically” and such. Overall, language quality is too low for the manuscript to get looked at for a publication in Cells in the current form.

Author Response

Dear Reviewer

Thank you very much for kindly reviewing our paper, titled as “The dynamic poly(A) tail acts as a signal hub in mRNA metabolism” (cells-2021669). We are delighted that you find the paper of interest and thank you for insightful and constructive comments. We have completely revised our manuscript and requested language polishing service from the linguistic expert in MDPI. Other specific points  you raised are answered as follows:

  1. In last submission our abstract was not well summarized to make it look similar to the introduction, however, our abstract has been thoroughly revised in the revised version and will not look the same as the introduction.

In this manuscript we are trying to make a point, poly (A) tail act as a signal hub. It is supervised by diverse polyadenylation /deadenylation mechanisms and emit signals rely on binding different proteins, exactly as a flagman commanded by a captain and sending signals through waving different flags. So we still think flagman is an appropriate metaphor. We would like to keep it. But we removed some confusing words such as “commanders” and “semaphores” in the abstract.

  1. “Automatically, a vague notion of that poly (A) tail might be a kind of signal generated” has been changed to “Researchers also developed a vague notion that the poly(A) tail might be a kind of generated signal”
  2. We have divided Figure 1 into three figures, A,B and C, and explained them separately in the revised version.
  3. “post-transcriptional and also non-templated addition”is a good description of the poly (A) tail, however it can also be described as “a modification of mRNA” just as Eckmann et al wrote “ Poly(A) tails have long been known as stable 3 modifications of eukaryotic mRNAs” in 2010 and Nicholson described “Poly(A) Tails Are a Dynamic and Important Modification of RNA”. (DOI: 10.1002/wrna.56 and DOI: 10.1016/j.tcb.2018.11.002)

We have change “polyadenylic acid” to “3’ end polyadenylic acid”

It should be admitted poly(A) stretches not only happened on 3’ end of mRNA, as these papers you listed we have seeing more roles of the A-rich sequences of mRNA. But in this manuscript we prefer to focus on 3'end mRNA poly(A) tail.

  1. The sub title on line 89 has been changed to “Nuclear polyadenylation regulates nascent pre-mRNA poly(A) tail generation”line 99 has changed to “PAS, an AAUAAA sequence located 15–30 nucleotide upstream of cleavage site on the 3’ untranslated region” 
  2. We have reorganized the paragraphs. In this section we intend to summarize the two steps (cleavage and polyadenylation) of the nuclear polyadenylation process. In first paragraph we listed all participant elements, complexes and enzyme, in second paragraph we described how cleavage happened on mRNA 3’end, in the third paragraph we portrayed what is canonical Poly(A) polymerase. Correspondingly, the introduce of PAPs are supposed not that abrupt.
  3. Cytoplasmic deadenylation causes the poly (A) tail shrinking, however a shorter tail does not always promote mRNA decay, in some cases A shortened poly (A) tail may also facilitate translation, thus we though 'regulate' is accurate and we would like to Maintain the title of 2.2 as “Cytoplasmic deadenylation regulates poly(A) tail shrinkage and mRNA decay”
  4. The sentence on 140 has been changed to “Nuclear polyadenylation promotes poly(A) tail extension. In contrast, cytoplasmic deadenylation contributes to poly(A) tail shrinkage. ”
  5. 2 is an independent sub-section which describe specific cytoplasmic deadenylation mechanism. This mechanism currently includes two complex-mediated shortening, PAN2-PAN3 and CCR4-NOT. We have introduce CCR4-NOT at 3rd   sentence as “PAN2–PAN3, CCR4–NOT, and PABPC are all widely accepted as playing vital roles in mediating poly(A) deadenylation”. It is therefore appropriate to expansion describe CCR4–NOTin line 161.
  6. 3 should be classified as a separate paragraph, because poly/de-adenylationdescribe in this paragraph are limited in oocyte or early embryonic cell. During this period, the variation of poly (a) tail length of mRNA very flexible. It can vary between polyadenylation and deadenylation to sculpture transcriptome according developmental demand.  
  7. We have changed the title into “Semaphore” poly(A) hub transmission with/without “flags”, which we think is closer to what we want to express. We are trying to make a point that poly(A) tail can metaphorically as a “flagman” send out signals with/without “flags” to regulate the orderly functioning of mRNA metabolism.
  8. We have modified Table 1 to make our meaning clearer and to cite the respective key references. We think that signal is originally used to adapt to the environment, signal is the adaptation to the dynamic environment, so the signal and the circumstance, the condition itself is inseparable. To our knowledge, the “poly(A) binding proteins”  don’t contribute to the signals directly, and more research is needed on the exact mechanism. What we know so far is that poly(A) tails and poly(A) binding proteins work together to transmit signals that regulate mRNA behavior. The difference of poly(A) tails and poly(A) binding proteins between species was larger than that between nuclear and cytoplasmic processes, so we chose species classification. It may be because the current research on polyA tail is still at a relatively preliminary stage, and many specific mechanisms and details have not been thoroughly studied. Perhaps after further research, we can reach more essential conclusions across the gap between species. We think the content of Table 1 is a little complicated to draw illustrations, which is not as clear as the table, so we did not draw illustrations.
  9. We've reorganized our language to be more refined in what we want to say. The third paragraph mainly aims to express the signals conveyed by the poly(A) tails, which are summarized in Table 1. We thought that drawing illustrations might make the problem more complicated, so we did not draw illustrations.
  10. We didn't mean that, maybe it was the language problem that caused the misunderstanding, we've reorganized our language to be more refined in what we want to say. We have deleted the misleading description and changed it to “The half-life of mRNA is correlated with the deadenylation rate rather than with tail length, we contend that the rate of deadenylation plays an important role in regulating the rate of translation[9,13].”
  11. It is possible that a language problem caused confusion in understanding the logic, and we have modified the language to make sure that our meaning is clearer. The newly modified section is described as “Deadenylation and decay rates have previously been studied only as derivations of studies of mRNA deadenylation and degradation dynamics [51,130-132]. Recent studies, however, have demonstrated that the deadenylation and decay rate constants, for which the ranges are numbered in thousands, are much greater than expected, and correspond to cytoplasmic lifetimes [13]. We therefore speculate that the rates of deadenylation and decay may serve as a signaling mode in cells, and that it they are broadly involved in mRNA metabolism.”
  12. We have downloaded and carefully read the literature you recommended, among which two articles PMIDs 34385261 and 30006991 have been inserted into the paper by us. The rest of the literature, although very good, is not consistent with our focus, so it is not cited in this paper.
  13. We are also aware of our deficiency in English writing. We have sought professional language polishing service and made a lot of modifications in the revised version

Sincerely

QuanWang and co-authors

Reviewer 2 Report

This review manuscript by the authors Zhang et al., covers the functional highlights of the poly(A) tail in very good detail. However, the text is in need of a LOT of editing for English grammar and language. English is a difficult language, but this manuscript is not yet in a form that is comprehensible. I have indicated some of the language errors in my Specific Comments, below, but the Authors should work with an English-language editor to fix all the errors.

There are many things I like about this manuscript. I like the Authors' sections on cytoplasmic deadenylation and on the roles of CPEBP and nuclear poly(A) tail length as an important mRNA export signal. Figure 1 is beautiful, and is a great summary of the roles of poly(A) tails (but, see my comment in Specific Comments, below). Please express compliments to the artist from this Reviewer.

I also like the Authors' metaphor of the poly(A) tail as a semaphore or flag. However, I would like the Authors to be more explicit about how they interpret the metaphor: Which part of the poly(A) tail is the "flag" (its presence on mRNAs? Its length? The proteins that bind to it?)? What are the different "flags" it waves? How does the "flagman" detect "signals" from the nucleus, through nuclear-cytoplasmic transport, or in the cytoplasm? A "flagman" implies a decision that must be made about the mRNA fate (not a decision by a human participant, but a decision nevertheless). How does that decision occur?

Specific Comments:

1. Pages 1-9 -- The Authors alternate between "poly (A)" (with a space) and "poly(A)" (without a space) throughout the manuscript. Please change all instances to "poly(A)" without the space.

2. Page 1, line 19 -- Please change "it's" to "its."

3. Page 1, line 22 -- Please change "sinals" to "signals."

4. Page 2, line 56 -- Please explain what is meant by "'commanders'".

5. Page3, lines 65-84 and page 4, lines 85-88 -- Is this whole paragraph meant to be the legend to Figure 1? It seems unusually long for a figure legend. I recommend reducing the length of the Figure 1 legend to a description of each of the parts of the figure, and inserting the additional information from the legend into the main text of the article.

6. Page 3, line 69 -- Please spell out the abbreviation "cPAP" in the figure legend.

7. Page 4, line 97 -- The Authors mention canonical poly(A) polymerase (cPAP) here, but do not define it until lines 131-133. I recommend moving the definition to line 97.

8. Page 5, line 174-175 -- Please change "PABC binded poly(A) tail deadenylate, while CAF1 catalyzes naked poly(A) tail deadenylate" to "PABC-bound poly(A) tail deadenylation, while CAF1 catalyzes naked poly(A) tail deadenylation."

9. Page 5, line 183 -- Please change "serum responses" to "serum response."

10. Page 6, line 225 -- Please change {3.“. Semaphores” poly(A)hub transmit with/with out “flags} to {3. "Semaphores” in the  poly(A) hub transmit signals with and without “flags"} (or, re-write if that is not what was meant).

11. Page 6, lines 225-231 -- I think I like the metaphor of the semaphores and flagmen with regard to poly(A) tail functions, but I am confused about the details. I would like a deeper explanation of this metaphor. What aspects of the poly(A) tail correspond to the semaphore? What aspects of the poly(A) tail correspond to the "flagman"? Are the semaphore and the flagman the same thing? What parts of mRNA metabolism (transcription, mRNA processing, transport, translation, etc.) are receiving the signals? If the poly(A) tail is a "hub," what kinds of functions ("spokes") emerge from the hub? For the hub metaphor, perhaps there would be a way to indicate it in Figure 1.

12. Page 6, line 232 (Table legend) -- Please give more explanation of the "signals," how they are "provided," and what the table tells the reader about these?

13. Page 7 (Table 1) -- Please give manuscript citations for each of the points (Poly(A) tail length, etc.) shown in this table.

14. Page 7 (Table 1) -- The Authors indicate that some (it is not clear) mRNAs have poly(A) tails in the nucleus. I am not aware of this (although I am not an expert). Please describe this value better.

15. Page 7 (Table 1) -- Please change "Asppergillus" to "Aspergillus."

16. Page 7, lines 244, 245, and 247 -- Please describe what are "U snRNAs" (probably the U1, U2, U4, and U5 snRNAs).

17. Page 8, line 270 -- Please italicize "Saccharomyces cerevisiae."

18. Page 9, line 339 -- Please italicize "Aspergillus nidulans."

19. Page 9, line 364-368 (Conclusions) -- In the Conclusions, the Authors state that "Poly (A) tail acts more like a dynamic signal hub than 367 an independent signal symbol." Please describe how this fits with their metaphor of the semaphore/flagman.

20. Page 9, lines 369-370 -- I do not understand "…mediate mRNA metabolize orderly." Please re-write for clarity.

Author Response

Dear Reviewer

Thank you very much for kindly reviewing our paper, titled as “The dynamic poly(A) tail acts as a signal hub in mRNA metabolism” (cells-2021669). We are delighted that you find the paper of interest and thank you for insightful and constructive comments. We have completely revised our manuscript and requested language polishing service from the linguistic expert in MDPI.

For the main points you raised, we have the following explanation:

In our metaphor we're trying to say whole poly(A) is act as a “flagman” which is regulate by polyadenylation and deadenylation, while diverse poly(A)-binding proteins (PABPN1, Nab2, PABPs, PAPBC1, Pab1, CAF1 and TUT4/7) can be seen as its “flags”. The “flagman” transmit various signals via bind different “flag” and also through different self lengths. Besides, You raised “How does the "flagman" detect "signals" from the nucleus, through nuclear-cytoplasmic transport, or in the cytoplasm?” while in our point of view the "flagman" does not detect "signals", it simply relays the information given to it by its superiors, just as in a fleet the captain made sailing adjustments based on changed navigation environment while this “order” is delivered to other ships by the “flagman”. Obviously, there supposed to have higher mechanism which is responsible in detecting circumstances and development in living organisms. Poly(A) tail just passing on some part of the instruction to the mRNA and determining what happens to the mRNA afterwards. For instance, When the poly (A) tail is extended and binds to PABPN1, this mRNA is more likely to export from nucleus rather than degraded.

Other specific points you raised are answered as follows:

  1. The space between poly and (A) are deleted in revised version.
  2. Change accepted.
  3. Change accepted.
  4. We instead of commanders with regulators.
  5. We have reorganized Figure 1 and rewritten the figure legend.
  6. Accepted.
  7. In this section we intend to summarize the two steps (cleavage and polyadenylation) of the nuclear polyadenylation process. In first paragraph we listed all participant elements, complexes and enzyme, in second paragraph we described how cleavage happened on mRNA 3’end, in the third paragraph we portrayed what is canonical Poly(A) polymerase and polyadenylation process. Thus, it is more convenient to introduce the nuclear polyadenylation mechanism without describing cPAPs in line 97 until line 131-133 in our point of view.
  8. Change accepted.
  9. Change accepted.
  10. We have rewrite this sentence “ Signals poly(A) hub transmission with/without “flags”
  11. As we answered in main point we're trying to say whole poly(A) is act as a “flagman”which is regulate by polyadenylation and deadenylation, while diverse poly(A)-binding proteins (PABPN1, Nab2, PABPs, PAPBC1, Pab1, CAF1 and TUT4/7) can be seen as its “flags”. The “flagman” transmit various signals via bind different “flag” and also through different self lengths. Besides, we deleted word“semaphores”.
  12. For this problem, our current research may only be at the stage of describing the phenomenon, and the deep mechanism is still unclear. What we do know is that the poly(A) tails is a dynamic “flagman”, the poly(A) tails binding protein is the “flag”, and they together make up the signal to the cell.
  13. We have modified Table 1 to make our meaning clearer and to cite the respective key references.
  14. Yes, nuclear polyadenylation occurs in the nucleus. Usually, only when a mRNA are polyadenylated and bind by PABPN1 it can be exported from nucleus. And we reorganized our table1.  
  15. We have made modifications according to your suggestion.
  16. The U snRNA here includes a variety of U snRNA types, including U1, U2, U4, and U5 snRNAs. Our citation is from article “Role of poly (A) tail as an identity element for mRNA nuclear export”.
  17. We have made modifications according to your suggestion.
  18. We have made modifications according to your suggestion.

Sincerely

QuanWang and co-authors

Reviewer 3 Report

The text is extremely difficult to read because the English language is poor. A detailed evaluation of the review/opinion paper will only be possible when a professionally revised version is available. Overall, the authors must clearly distinguish between effects on the poly(A) tails and effects through the poly(a) tails when describing the processes and interactions.  In some places, the language is too flowery and raises expectations that the manuscript does not fulfil, because it essentially describes mechanisms that regulate the length of the poly(a) tails, but not the effect of the poly( a) tails on the transcripts.

The authors have done an enormous amount of work in compiling the various interactions described in the literature and have used more than 100 references, which must be acknowledged. Each section should, however, begin with a clear assignment of the functions described below and end with an appropriate summary that gives the reader a clear orientation. 

Author Response

Dear review

Thank you very much for your comments and suggestions. We have realized our shortcomings in language, and we have sought the help from language experts. A revised version will be submitted and we look forward to your constructive comments.

Sincerely

QuanWang and co-authors

Round 2

Reviewer 1 Report

The authors seem to have addressed all or most of the comments and the manuscript in on the good path to improvement. I liked the figure updates and de-complication of the language, which now is more acceptably scientific.

I appreciate that the authors provided a version with all changes highlighted, this helps to immediately grasp what was altered. However, the tracked changes are in the PDF format and are obstructing understanding of the final product, sometimes very substantially, which hampers the usefulness of feedback.Table is nearly impossible to interpret as a result of this, for example.

I could still spot some expressions and language bits that could be recommended for improvement, but to fully address this and provide a recommendation could the authors please also supply a version with all changes accepted?

Reviewer 2 Report

This review has been greatly improved by careful editing and re-writing of relevant sections. Good job!